# Trends in Health-Risk Behaviors and Psychological Distress among Australian First-Year University Students

**DOI:** 10.3390/ijerph21050620

**Published:** 2024-05-14

**Authors:** Alison Knapp, Tracy Burrows, Megan Whatnall, Lucy Leigh, Sarah Leask, Melinda Hutchesson

**Affiliations:** 1School of Health Sciences, College of Health, Medicine and Wellbeing, University of Newcastle, Callaghan, NSW 2308, Australia; alison.knapp@uon.edu.au (A.K.); tracy.burrows@newcastle.edu.au (T.B.); megan.whatnall@newcastle.edu.au (M.W.); 2Food and Nutrition Research Program, Hunter Medical Research Institute, New Lambton Heights, NSW 2305, Australia; 3Data Sciences, Hunter Medical Research Institute, New Lambton Heights, NSW 2305, Australia; lucy.leigh@hmri.org.au (L.L.); sarah.leask@hmri.org.au (S.L.)

**Keywords:** health behaviors, psychological distress, university students, mental health

## Abstract

University students are recognized as a high-risk population group who experience greater rates of poor health outcomes and mental ill-health. Commencing university is recognized as a major life transition, where students experience new financial, academic, environmental, and social pressures that can cause changes in their normal behaviors. This study explored trends in health-risk behaviors and psychological distress in commencing university students over four survey years. First-year undergraduate students, aged 17–24, from an Australian university were included. A secondary analysis was performed on data collected via cross-sectional surveys on four occasions (2016, 2017, 2019, 2020). Crude logistic regression models were utilized to investigate the association between meeting guidelines and survey year. Odds ratios for the pairwise comparison between each year are reported. In this analysis, 1300 (2016), 484 (2017), 456 (2019), and 571 (2020) students were included. Analyses showed two clear trends: students’ probability of being at high/very high risk of psychological distress (35–55%) and consuming breakfast daily (44–55%) consistently worsened over the four survey years. These findings suggest that the odds of psychological distress and daily breakfast consumption worsened over time, whilst the proportion of students engaging in some health-risk behaviors was high, highlighting the importance of early intervention during the transition to university.

## 1. Introduction

The mental health of commencing university students is of concern. The World Health Organization (WHO) World Mental Health International College Student (WMH-ICS) initiative reported that one-third of students in their first year of university screened positive for at least one mental disorder, including major depression and generalized anxiety disorder [1]. It is also acknowledged the university student population has high rates of health-related risk behaviors [2]. For example, the National College Health Assessment (NCHA) surveys in the USA (2022, *n* = 50,307) found that 83% of students consumed less than three servings of vegetables per day, 60% did not meet physical activity guidelines, 43% slept less than recommended, 14% were smokers, 19% consumed five or more standard drinks in one sitting, and 14% recently used illicit drugs [3]. Health-risk behaviors often do not occur in isolation; rather, students engage in multiple health-risk behaviors [2,4].

The increased prevalence of health-risk behaviors and poor mental health among commencing university students is of concern due to the significant impact on long-term health, quality of life, and academic careers [5]. For example, poor mental health among commencing students has been associated with reduced university attendance and performance, higher rates of withdrawal from university, and lower rates of future employment [6].

In 2020, there were over 1.6 million students enrolled at Australian universities, with more than one-third being commencing students [7]. Young adults aged 17–25 years typically undergo a significant life transition [8], and for many, this coincides with leaving high school and entering university for the first time. This transition has been suggested as an important determinant of health-risk behaviors and mental health status [9], where first-year students are faced with greater responsibilities, often living independently for the first time, experience financial and academic pressures, and their usual behaviors are challenged by personal, environmental, and social factors [10]. Commonly, physical activity decreases, sedentary behaviors increase, significant changes in diet often occur [9], sleep quality is poor, and the use of alcohol, drugs, and smoking increases [4]. This critical period can influence the trajectory of health-risk behaviors and mental health into later life, impacting their overall wellbeing [11].

Whilst the transition to university is significant, the emergence of COVID-19 further impacted the wellbeing of university students. Research has suggested an increased prevalence of psychological stress, depression, and anxiety and changes in health behaviors among students in response to deviations from their normal routines due to COVID-19 [12]. For example, a meta-analysis of 20 studies (73,912 participants) found the prevalence of depression among college students during the pandemic was 39% [12], which is higher compared to the global prevalence of 4.4% [13]. Further, a systematic review of seven studies indicated most reported a significant increase in sedentary time and a decrease in light-intensity physical activity among undergraduate students and not graduate students during the pandemic [14].

There is a limited number of studies that have considered how health-risk behaviors and psychological distress are changing among young adults (aged 17–24) commencing university. To our knowledge, no studies have tracked changes in a broad range of health-risk behaviors and psychological distress over time amongst a sample of young adult commencing university students, particularly prior to and during the COVID-19 pandemic. Therefore, this study aims to explore trends in health-risk behaviors (poor diet, physical inactivity, sedentary behaviors, alcohol intake, smoking, drug use, and poor sleep) and psychological distress among commencing Australian university students, aged 17 to 24 years, over four survey years—from 2016 to 2020.

## 2. Materials and Methods

### 2.1. Study Design

We conducted a secondary analysis of data collected in cross-sectional surveys (University of Newcastle (UON) Student Healthy Lifestyle Survey [SHLS]) [15] over four years (2016, 2017, 2019, 2020). From 2017, the survey was to be repeated every second year; however, due to the COVID-19 pandemic, the survey was also distributed in 2020. An overview of the SHLS methodology is summarized in Table 1. The SHLS was conducted online through “Survey Monkey” for the years 2016, 2017, and 2019, and using “Blue” for 2020, and it took approximately 15 min to complete. These survey tools allowed students to only access the survey once on a single device, to prevent multiple entries by the same student. This analysis includes health-risk behavior and psychological distress data. Approval for this study was received from the UON Human Research Ethics Committee (H-2015-0459).

### 2.2. Participants and Recruitment

All students enrolled at the UON at the time of the survey were invited to participate (2016 *n* = 30,053, 2017 *n* = 33,744, 2019 *n* = 34,924, 2020 *n* = 37,249); however, this analysis only includes responses from participants who indicated they were in their first year of study, in an undergraduate program, and aged 17–24 years. The UON is a large, regional university with students based at the main campus located in Callaghan, New South Wales (NSW), Australia, plus other smaller campuses across NSW (*n* = 4). Students were invited to participate via email to their UON email account, with two reminder emails sent. The survey was advertised through student social media accounts, on-campus computer screens, and posters, and UON staff were asked to promote the survey in class. Upon survey completion, students were eligible to enter a prize draw for gift vouchers valued at AUD 100-AUD 250. All students provided written consent to participate in the survey, and all answers were submitted anonymously so no student could be identified via their answers.

### 2.3. Measures

#### 2.3.1. Demographic and Student-Related Characteristics

Demographic data were collected across the four surveys based on questions adapted from the National Census [16] to ensure suitability for a university student population. This included age, gender (male, female, non-binary, or another gender identity), Aboriginal or Torres Strait Islander descent, language, marital status, hours of paid work per week, financial support, and living situation. Student-related characteristics were also captured, including domestic/international enrolment, campus location, and the faculty of study.

#### 2.3.2. Health-Risk Behaviors and Psychological Distress

Diet: This was assessed using short diet questions [17]. Fruit and vegetable intakes were assessed as usual servings/day, ranging from “I don’t eat fruit [vegetables] to 6 or more serves” [17]. One serving of fruit was defined as 150 g, and one serving of vegetables was 75 g [18]. Fruit and vegetable intake is reported as a percentage of consuming the recommended amount (>2 for fruit and >5 or 5.5 servings/day for vegetables) based on age and gender [18]. The frequency of consumption of fast foods was assessed with responses ranging from “never or rarely” to “everyday”. A higher consumption of fast foods was considered as one or more times/week, as per the Australian Guide to Healthy Eating (AGHE) lower end of the range for discretionary items [18]. Frequency of breakfast consumption was assessed with responses ranging from “never or rarely” to “everyday”, with responses categorized as consuming breakfast every day or less than every day.

Physical activity: This was assessed via the Active Australia Survey [19], including the number of sessions and total time (minutes) performing moderate, vigorous, and walking activity in the previous week [19]. The total number of sessions of physical activity, total time spent in each activity, and the proportion of students participating in a sufficient amount of activity to gain health benefits, as per national recommendations (i.e.,>150 min moderate activity/week over >5 sessions) [20], were determined.

Sitting time: This was assessed using questions from the NSW Adult Population Health Survey [17], including average time spent sitting on a weekday and on a weekend day. Average total sitting time was then calculated ((total time spent sitting on a weekday × 5) + (total time spent sitting on a weekend day × 2)/7)). Participants were categorized as meeting recommendations if they sat for <8 h/day, based on the evidence that there is a greater mortality risk for increased sitting time, in comparison with <8 h/day [21].

Tobacco smoking/e-cigarettes: Participants’ tobacco smoking status was assessed, with those reporting they smoke “daily” or “occasionally” classified as smokers. All others were deemed non-smokers [17].

Alcohol: Intake was assessed using frequency of consumption and quantity, i.e., the number of standard drinks usually consumed per drinking occasion, with one standard drink defined as 10 g of pure alcohol [22]. The quantity of alcohol consumed was compared with national recommendations for single-occasion risk (>4 standard drinks/single occasion) [23].

Illicit drug use: This was assessed by determining whether participants had used non-prescription drugs during their lifetime, using questions adapted from the National Drug Strategy Household Survey [24].

Sleep: This was assessed using one question from the National Centre for Chronic Disease Prevention and Health Promotion [25]. Participants indicated their average hours of sleep in a 24 h period and were classified as either meeting/not meeting the Sleep Health Foundation age-based recommendations [26] (8–10 h for 17-year-olds and 7–9 h for 18–64-year-olds) [26].

Psychological distress: This was assessed using the Kessler Psychological Distress Scale (K-10) questionnaire [27]. Scores for each of the 10 items were summed, and the level of severity of nonspecific psychological distress was categorized: low to moderate risk (score of 21 or less) or high to very high risk (score of greater than 21) [27].

### 2.4. Statistical Analysis

Descriptive statistics for demographics included frequencies for each variable. Crude logistic regression models were utilized to investigate the association between meeting guidelines and survey year (for each of the health-risk behaviors and psychological distress outcomes). Odd ratios and 95% confidence intervals for the pairwise comparisons between each year are reported. Each of the eleven models was then adjusted in a multivariable fashion for age, gender, domestic/international student, and Indigenous status. All statistical analyses were programmed using SAS v9.4 (SAS Institute, Cary, CA, USA).

## 3. Results

### 3.1. Participant Characteristics

Across the four surveys, 1300 (2016), 484 (2017), 456 (2019) and 571 (2020) students were included (Figure 1). Student characteristics are summarized in Table 2. Over the four surveys, most of the sample were domestic students (91–96% (range of responses across four surveys)), female (70–71%), and aged 19 (22–39%), and up to 3.6% identified as Aboriginal.

### 3.2. Health-Risk Behaviors

The proportion of students meeting and not meeting the guidelines for each of the health-risk behaviors and mental health outcomes for each of the survey years is shown in Figure 2 and Appendix A, and pairwise comparisons of student health-risk behaviors and mental health results are shown in Table 3.

#### 3.2.1. Diet

The number of students meeting the recommended servings of fruit per day ranged from 49 to 57% across the four surveys. The adjusted model indicates that students in 2016 had a higher probability (36% increase in odds) of meeting fruit recommendations than students in 2019, and students in 2017 were 31% more likely than students in 2019.

Participants meeting vegetable recommendations were similar across the four surveys, ranging from 6.6 to 8.6%.

Participants who reported consuming breakfast every day ranged from 44 to 55% across the four surveys. There was a consistent decrease over time: 55% in 2016, 54% in 2017, 48% in 2019, and 44% in 2020. There was a 25% increase in the odds of students in 2016 consuming breakfast daily compared to students in 2017 and a 48% increase in the odds of students consuming breakfast in 2016 than students in 2020, and students were 46% more likely in 2017 than students in 2020 to consume breakfast.

Between 57 and 64% of participants reported consuming fast food <1–2 times per week over the four surveys. There was an approximate 34% increase in the odds of students in 2016 consuming fast food <1–2 times per week than a student in 2020, and students in 2019 had a higher probability (30% increase in odds) than students in 2020.

#### 3.2.2. Physical Activity

Participants reporting performing sufficient physical activity per week to meet national recommendations ranged from 67 to 77% across the four surveys. There was a 48% increase in the odds of students in 2016 meeting physical activity recommendations compared to students in 2017, and students in 2017 were 40% less likely to perform sufficient physical activity than students in 2020.

#### 3.2.3. Sitting Time

Students reporting sitting <8 h/day across the four surveys ranged from 57 to 67%. There was a 32% increase in the odds of students sitting <8 h/day than in 2017. In 2016, students were 32% more likely to sit <8 h/day than students were in 2017. In 2017, students were 37% less likely to sit <8 h/day than students in 2019 and students in 2017 were 27% less likely to sit for <8 h/day compared to students in 2020. 

#### 3.2.4. Smoking

Respondents reporting to be non-smokers ranged from 92 to 95% over the four surveys. Students in 2016 had a lower probability (47% decrease in odds) of being non-smokers than students in 2017, and there was a 97% increase in the odds of students in 2017 being non-smokers than students in 2019.

#### 3.2.5. Alcohol

Participants reporting a low risk of alcohol single-occasion risk ranged from 55 to 65% across the four surveys. In 2016, students were 36% more likely to engage in low-risk single-occasion alcohol consumption compared to students in 2017, and students in 2017 had a lower probability (32% decrease in odds) of engaging in single-occasion alcohol risk than students in 2020.

#### 3.2.6. Illicit Drug Use

Respondents over the four surveys reporting no use of non-prescription drugs during their lifetime ranged from 59 to 69%. Students in 2016 had a higher probability (40% increase in odds) of having no lifetime drug use than students in 2019.

#### 3.2.7. Sleep

Participants reporting to meet sleep recommendations ranged from 71 to 80% across the four surveys. There was a 37% decrease in the odds of students in 2016 meeting sleep recommendations than students in 2019 and a 29% decrease in the odds of students in 2017 than students in 2019, and students in 2019 were 43% more likely to meet sleep recommendations compared to students in 2020.

### 3.3. Psychological Distress

Participants categorized as being at high/very high psychological distress risk ranged from 35 to 55%. A consistent increase over the four surveys was shown, with 35% in 2016, 51% in 2017, 53% in 2019, and 55% in 2020. Comparatively, the number of students classified as low–moderate risk of psychological distress risk was highest in 2016 at 65%, followed by 49% in 2017, 47% in 2019, and 45% in 2020. Students in 2016 had a higher probability (91% increase in odds) of being at low–moderate risk of psychological distress than students in 2017; students in 2016 had a higher probability (112% increase in odds) than students in 2019; and students in 2016 had a higher probability (131% increase in odds) than students in 2020.

## 4. Discussion

This study explored differences in the odds of health-risk behaviors and psychological distress in young adults in their first year of university at a large regional Australian university over four surveys undertaken from 2016 to 2020. The findings of this study indicate the proportion of students engaging in some health-risk behaviors and experiencing psychological distress was high. Most notably, the odds of psychological distress risk and daily breakfast consumption worsened over time. For example, students in 2020 had a higher probability of being at high to very high risk of psychological distress and not consuming breakfast daily compared to students in 2016. The study’s findings may be used to support the need for early intervention during the transition young adults experience when entering university for the first time to address health-risk behaviors and psychological distress.

Psychological distress was prevalent among commencing university students over the four surveys, as assessed in the study using Kessler 10, which has been shown to be related to mental health diagnosis [27]. Trends over the four years saw the proportion of students being at low–moderate risk of psychological distress declining (65% in 2016 to 45% in 2020), with a subsequent increase in the proportion of students being at high–very high risk of psychological distress (35% in 2016 to 55% in 2020). A previously published analysis of the 2017 SHLS including all university students (*n* = 3077) reported that 37.6% of students were classed as being at high or very high risk of psychological distress [15]. Similarly, 20% of Australians aged 16–34 years experienced high or very high psychological distress in 2020–2021 [28]. Evidently, there is a large disparity between young adult students commencing university and all students and Australian population groups. This, therefore, further supports the importance of implementing interventions during the transition to university to improve students’ psychological wellbeing during their university studies.

The worsening of daily breakfast consumption over the four surveys was identified as the only clear trend occurring out of all the health-risk behaviors assessed. Students were more likely to consume breakfast daily in 2016, with a decrease of 11% by 2020. Research has previously identified potential barriers to students consuming regular breakfast, including financial insecurity, negative mood, health conditions, limited cooking facilities, weight control, and poor time management skills [29,30,31]. Whilst these barriers exist, the current study only identified the trend that students consume breakfast less than previously, and we did not explore the potential contributing factors concerning why this may be occurring among this population group or worsen over time. Therefore, further studies would benefit from investigating and linking the correlations between barriers of daily breakfast intake to help establish interventions to improve the dietary behaviors of commencing students.

A unique characteristic of this study was the ability to track health-risk behaviors and mental health throughout the COVID-19 pandemic, which provides insights into the potential effects COVID-19 had on young adults during their first year at university in 2020. Our findings suggest that the odds of several health-risk behaviors (breakfast and fast-food consumption, sitting time, and sleep) and psychological distress risk worsened between 2019 and 2020. These findings are similar to recent studies reporting first-year university students’ health behaviors and mental health changes during and after the emergence of COVID-19 compared to pre-pandemic [32,33]. Undergraduate students may have been more affected by online modifications to teaching and social isolation due to less resilience (compared to other graduates) in their adaption to changes in their normal routines [34], subsequently resulting in alterations to their eating, sleeping, and sedentary behaviors and mental state [12,14,32,33,35]. It should be acknowledged that the current study did not explore the direct impact that COVID-19 had on students’ health and wellbeing, and so the change in health behaviors and psychological distress from 2019 to 2020 cannot be explained directly by the pandemic.

Whilst notable trends in other health-risk behaviors over the four years were not shown, the rates of some health-risk behaviors remained consistently poor over time and reflect similar findings from other studies [2,4,5,9,15]. For example, fruit and vegetable intake among the current study’s participants coincides with the trend of poor intake among young adults and university students. For instance, a study conducted on Australian university students found that only 10.5% of students met vegetable recommendations and 50.7% met fruit recommendations [15]. Many students in the current study across the four years surveyed undertook risky behaviors, including alcohol use, where 35–39% of participants exceeded the alcohol single-occasion risk (i.e., more than four standard drinks/day) [23]. This is almost double when compared to Australians aged 14 and older (17.2%) [36], and the American College Health Assessment (2023) reports that 20.8% of students consume five or more drinks/occasion [37]. These findings also support the notion that the transitional period first-year university students undergo may significantly impact their health behaviors when compared to other university cohorts [10].

The rates of health-risk behaviors among the current study participants may be lower than those of the general population for physical activity and smoking. For example, 23–33% of participants reported not performing sufficient physical activity; this is similar to 29.7% of American college students [37], and there is a large difference when compared to 77.6% of Australian adults not meeting recommendations [38]. Similarly, the proportion of participants in this current study reporting to be smokers ranged from 4.5 to 8.1%, which is approximately half when compared to Australian adults (10.6%) and far less than 21.2% of American college students [37]. The proportion of students reporting lower rates of health-risk behavior engagement may be attributed to several factors, including the type of degree students are enrolled in, UON being a smoke-free campus, and increased initiatives nationwide to reduce smoking prevalence [39].

Some limitations of this study include the use of a cross-sectional design and self-reported data. However, the use of validated tools such as the Kessler Psychological Distress Scale reduces the potential bias from self-reporting. There were small variances among the four surveys, including the questions asked, the data collection period differing across the surveys, and data not being collected in 2018. It is acknowledged that the use of a convenience sample and some student sub-groups (e.g., male students) being underrepresented could influence the results by underestimating the level of some health-risk behaviors and psychological distress and not reflecting the whole university population. Additionally, the study did not explore the impact that individual factors (i.e., gender) may have had on health-risk behaviors or psychological distress outcomes or the association between the different health-risk behaviors, highlighting key areas future research should explore. The strengths of this study include the large sample size and the inclusion of four surveys in the unique data set, which enabled the trends to be tracked over time, including pre- and mid-COVID-19 pandemic.

## 5. Conclusions

The findings from this cross-sectional analysis highlight that first-year young adult university students are engaging in health-risk behaviors and experiencing worsening psychological distress. Given that the transitional period from high school to university has been deemed an important determinant in the health of university students at enrolment, interventions should be targeted within secondary education settings and follow on into tertiary education settings. Future studies should include cohort studies to track changes in health-risk behaviors and psychological distress in first-year university students over time prospectively and aim to study the associations between students entering university.

## Figures and Tables

**Figure 1 ijerph-21-00620-f001:**
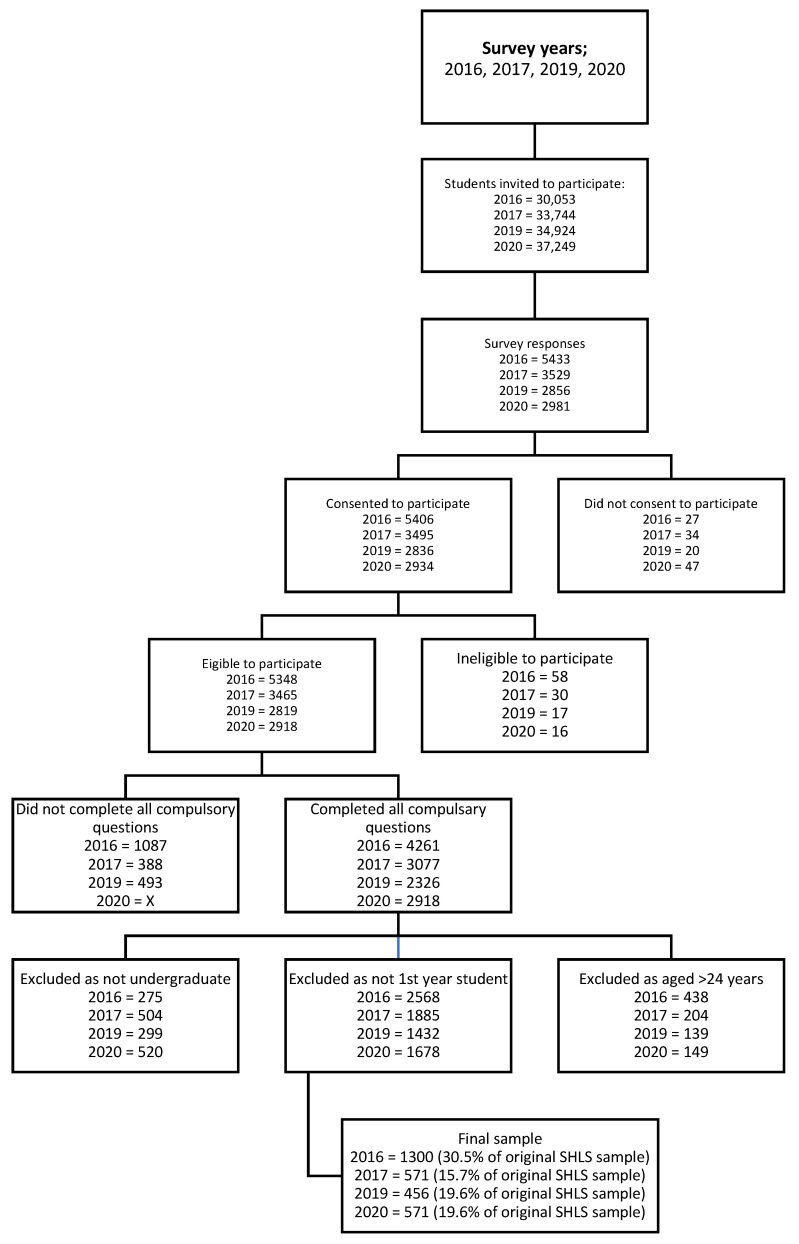
Flow diagram of Student Healthy Lifestyle Survey (SHLS) respondents from years 2016, 2017, 2019 and 2020.

**Figure 2 ijerph-21-00620-f002:**
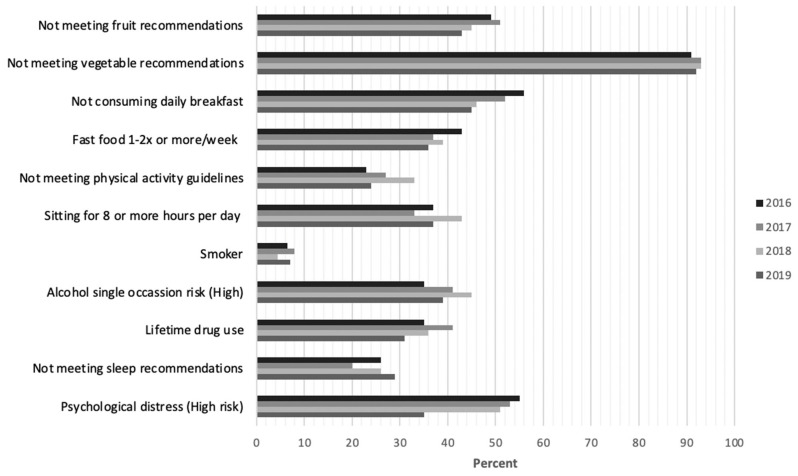
The proportion of first-year UON students, in an undergraduate program, aged 17–24 years, not meeting the guidelines/recommendations for each individual health-risk behavior (including diet, physical activity, sitting time, smoker, alcohol and drug use and sleep) and psychological distress outcomes, for each “Student Healthy Lifestyle Survey” (SHLS) year (2016, 2017, 2019, 2020).

**Table 1 ijerph-21-00620-t001:** Overview of recruitment, timeframes, and outcomes for the Student Healthy Lifestyle Survey (SHLS) in years 2016, 2017, 2019 and 2020.

Survey Year	2016	2017	2019	2020
Eligibility	All UON students (*n* = 30,053)	All UON students (*n* = 33,744)	All UON students (*n* = 34,924)	All UON students (*n* = 37,249)
Recruitment time frame	15 February 2016–13 March 2016 During orientation and first 3 weeks of semester 1	4 September 2017–1 October 2017 During semester 2	9 September 2019–5 October 2019 During semester 2	6 October 2020–November 2020 During semester 2
Invitation methods	Email invitation sent 15 February 2016 2 reminder/thankyou emails sent to students 1 and 2 weeks after the original email Incentive: 1 in 10 prizes valued at up to $250	Email invitation sent 4 September 2017 2 reminder/thankyou emails sent to students 1 and 2 weeks after the original email Incentive; 1 in 5 $100 gift vouchers	Email invitation sent 9 September 2019 2 reminder/thankyou emails sent 1 and 2 weeks after the original email Incentive: 1 in 5 $100 gift vouchers	Email invitation sent 6 October 2020 2 reminder/thankyou emails sent 1 and 2 weeks after the original email Incentive; 1 in 10 $100 gift cards
Advertisement	Advertised over 4 weeks; -Social media accounts -Screen savers on all university computers -Posters displayed on noticeboards -Lecturers advertised at the beginning of class using a PowerPoint slide	Advertised over 4 weeks; -Social media accounts -Screen savers on all university computers -Posters displayed on noticeboards -Lecturers advertised at the beginning of class using a PowerPoint slide	Advertised over 4 weeks; -Social media accounts -Screen savers on all university computers -Posters displayed on noticeboards -Lecturers advertised at the beginning of class using a PowerPoint slide	Advertised over 4 weeks; -Social media accounts -Screen savers on all university computers -Lecturers advertised at the beginning of class using a PowerPoint slide
Study design	-Online cross-sectional survey via survey monkey -Anonymous answers -Provided consent -Could only submit one entry -15 min to complete	-Online cross-sectional survey via survey monkey -Anonymous answers -Provided consent -Could only submit one entry -15 min to complete	-Online cross-sectional survey via survey monkey -Anonymous answers -Provided consent -Could only submit one entry -15 min to complete	-Online cross-sectional survey via Blue -Anonymous answers -Could only submit one entry -Provided consent -15 min to complete
Total questions	66 over 30 pages	61 over 27 pages	71 over 34 pages	50 questions
Sample size who completed all compulsory survey questions (*n*) (%)	*n* = 4261 (14.2%)	*n* = 3077 (9.1%)	*n* = 2326 (6.7%)	*n* = 2918 (7.8%)
Total who answered Non-compulsory questions (*n*) (%)	4095 drug use (13.6%) 4134 Mental health and wellbeing (13.7%)	3043 Other drugs (9%) 3003 mental health (8.9%)	2142 other drugs (6.1%) 2265 mental health (6.5%)	2709 Other drugs (7.6%) 2816 Mental health (7.5%)

**Table 2 ijerph-21-00620-t002:** Characteristics of first-year UON students, in an undergraduate program, aged 17–24 years, who completed the Student Healthy Lifestyle Survey (SHLS) in years 2016, 2017, 2019, and 2020.

Characteristic		2016 (*n* = 1300)	2017 (*n* = 484)	2019 (*n* = 456)	2020 (*n* = 571)
Domestic/International	Domestic	1253 (96%)	456 (94%)	438 (96%)	520 (91%)
	International (onshore)	29 (2.2%)	22 (4.5%)	13 (2.9%)	18 (3.2%)
	International (offshore)	18 (1.4%)	6 (1.2%)	5 (1.1%)	33 (5.8%)
Gender	Male	380 (29%)	136 (28%)	129 (28%)	168 (29%)
	Female	915 (70%)	345 (71%)	320 (70%)	399 (70%)
	Other	5 (0.4%)	3 (0.6%)	-	-
	Non-binary	-	-	3 (0.7%)	3 (0.5%)
	Another gender identity	-	-	4 (0.9%)	1 (0.2%)
Age (years)	17	90 (6.9%)	2 (0.4%)	1 (0.2%)	-
	18	527 (41%)	100 (21%)	106 (23%)	115 (20%)
	19	289 (22%)	158 (33%)	176 (39%)	212 (37%)
	20	143 (11%)	89 (18%)	70 (15%)	104 (18%)
	21	77 (5.9%)	51 (11%)	38 (8.3%)	47 (8.2%)
	22	54 (4.2%)	33 (6.8%)	32 (7.0%)	35 (6.1%)
	23	63 (4.8%)	24 (5.0%)	16 (3.5%)	33 (5.8%)
	24	57 (4.4%)	27 (5.6%)	17 (3.7%)	25 (4.4%)
Aboriginal or Torres Strait Islander Origin	Aboriginal	47 (3.6%)	13 (2.7%)	16 (3.5%)	15 (2.6%)
	Torres Strait Islander	-	-	1 (0.2%)	3 (0.5%)
	Aboriginal and Torres Strait Islander	2 (0.2%)	-	1 (0.2%)	-
	None of the above	1251 (96%)	471 (97%)	437 (96%)	541 (95%)
	No response			1 (0.2%)	12 (2.1%)

**Table 3 ijerph-21-00620-t003:** Pairwise comparisons of student health-risk behaviors and mental health results (first-year UON students, in an undergraduate program, aged 17–24 years) by year (2016, 2017, 2019, 2020)—crude and adjusted models (modeling the odds of “meeting guidelines”).

Health Behavior	Comparison Years	Crude Odds Ratio (95% CI)	Crude *p*-Value	Adjusted Odds Ratio (95% CI)	Adjusted *p*-Value
Meeting fruit recommendations	2016 vs. 2017	1.09 (0.89, 1.35)	0.400	1.04 (0.83, 1.29)	0.749
	2016 vs. 2019	1.37 (1.11, 1.70)	0.004	1.36 (1.09, 1.69)	0.007
	2016 vs. 2020	1.25 (1.03, 1.53)	0.024	1.18 (0.96, 1.45)	0.113
	2017 vs. 2019	1.25 (0.97, 1.62)	0.084	1.31 (1.01, 1.70)	0.043
	2017 vs. 2020	1.15 (0.90, 1.46)	0.269	1.14 (0.89, 1.46)	0.298
	2019 vs. 2020	0.91 (0.71, 1.17)	0.477	0.87 (0.68, 1.12)	0.281
Meeting vegetable recommendations	2016 vs. 2017	1.19 (0.79, 1.78)	0.405	1.24 (0.81, 1.88)	0.322
	2016 vs. 2019	1.23 (0.81, 1.88)	0.326	1.26 (0.82, 1.95)	0.292
	2016 vs. 2020	0.93 (0.65, 1.32)	0.673	0.90 (0.62, 1.30)	0.579
	2017 vs. 2019	1.04 (0.62, 1.73)	0.883	1.02 (0.61, 1.72)	0.936
	2017 vs. 2020	0.78 (0.49, 1.23)	0.287	0.73 (0.46, 1.16)	0.184
	2019 vs. 2020	0.75 (0.47, 1.20)	0.233	0.71 (0.44, 1.15)	0.168
Breakfast consumption (everyday)	2016 vs. 2017	1.06 (0.86, 1.31)	0.575	1.01 (0.82, 1.26)	0.901
	2016 vs. 2019	1.30 (1.05, 1.61)	0.016	1.25 (1.00, 1.55)	0.048
	2016 vs. 2020	1.57 (1.29, 1.91)	0.000	1.48 (1.20, 1.82)	0.000
	2017 vs. 2019	1.22 (0.95, 1.58)	0.122	1.23 (0.95, 1.59)	0.116
	2017 vs. 2020	1.48 (1.16, 1.89)	0.002	1.46 (1.14, 1.87)	0.003
	2019 vs. 2020	1.21 (0.94, 1.55)	0.135	1.19 (0.92, 1.52)	0.182
Fast food consumption (<1–2 times/week)	2016 vs. 2017	1.13 (0.91, 1.40)	0.268	1.15 (0.92, 1.43)	0.222
	2016 vs. 2019	1.04 (0.83, 1.29)	0.748	1.03 (0.82, 1.30)	0.796
	2016 vs. 2020	1.32 (1.08, 1.61)	0.007	1.34 (1.09, 1.66)	0.006
	2017 vs. 2019	0.92 (0.71, 1.20)	0.527	0.90 (0.69, 1.17)	0.430
	2017 vs. 2020	1.17 (0.91, 1.49)	0.222	1.17 (0.91, 1.50)	0.228
	2019 vs. 2020	1.27 (0.99, 1.63)	0.063	1.30 (1.00, 1.68)	0.046
Performing sufficient physical activity	2016 vs. 2017	1.56 (1.23, 1.97)	0.000	1.48 (1.17, 1.89)	0.001
	2016 vs. 2019	1.18 (0.92, 1.50)	0.199	1.14 (0.88, 1.47)	0.319
	2016 vs. 2020	0.94 (0.74, 1.20)	0.628	0.89 (0.69, 1.14)	0.337
	2017 vs. 2019	0.75 (0.57, 1.00)	0.054	0.77 (0.57, 1.03)	0.074
	2017 vs. 2020	0.61 (0.46, 0.80)	0.000	0.60 (0.45, 0.79)	0.000
	2019 vs. 2020	0.80 (0.60, 1.07)	0.137	0.78 (0.58, 1.05)	0.098
Sitting time (<8 h/day)	2016 vs. 2017	0.78 (0.63, 0.96)	0.021	1.32 (1.06, 1.64)	0.014
	2016 vs. 2019	1.23 (0.98, 1.54)	0.072	0.83 (0.66, 1.05)	0.118
	2016 vs. 2020	1.03 (0.84, 1.27)	0.747	0.97 (0.78, 1.20)	0.751
	2017 vs. 2019	1.58 (1.21, 2.06)	0.001	0.63 (0.48, 0.83)	0.001
	2017 vs. 2020	1.33 (1.04, 1.70)	0.025	0.73 (0.57, 0.94)	0.016
	2019 vs. 2020	0.84 (0.65, 1.09)	0.190	1.16 (0.89, 1.52)	0.264
Tobacco smoking (non-smoker)	2016 vs. 2017	0.63 (0.39, 1.01)	0.054	0.53 (0.33, 0.87)	0.011
	2016 vs. 2019	1.16 (0.78, 1.73)	0.465	1.05 (0.70, 1.58)	0.816
	2016 vs. 2020	0.91 (0.61, 1.35)	0.639	0.78 (0.51, 1.17)	0.229
	2017 vs. 2019	1.85 (1.08, 3.19)	0.026	1.97 (1.14, 3.41)	0.016
	2017 vs. 2020	1.46 (0.85, 2.50)	0.175	1.46 (0.84, 2.53)	0.182
	2019 vs. 2020	0.78 (0.49, 1.26)	0.315	0.74 (0.46, 1.20)	0.221
Low-risk alcohol consumption (single occasion)	2016 vs. 2017	1.29 (1.04, 1.59)	0.018	1.36 (1.09, 1.69)	0.006
	2016 vs. 2019	1.07 (0.86, 1.33)	0.520	1.10 (0.87, 1.37)	0.427
	2016 vs. 2020	0.85 (0.70, 1.05)	0.129	0.92 (0.74, 1.14)	0.455
	2017 vs. 2019	0.83 (0.64, 1.08)	0.168	0.81 (0.62, 1.05)	0.112
	2017 vs. 2020	0.66 (0.52, 0.85)	0.001	0.68 (0.53, 0.87)	0.003
	2019 vs. 2020	0.79 (0.62, 1.02)	0.076	0.84 (0.65, 1.09)	0.191
No lifetime drug use	2016 vs. 2017	1.24 (1.00, 1.55)	0.055	1.08 (0.85, 1.36)	0.528
	2016 vs. 2019	1.50 (1.20, 1.87)	0.000	1.40 (1.11, 1.76)	0.005
	2016 vs. 2020	1.16 (0.94, 1.44)	0.155	1.10 (0.88, 1.37)	0.399
	2017 vs. 2019	1.21 (0.93, 1.57)	0.164	1.30 (0.99, 1.70)	0.063
	2017 vs. 2020	0.94 (0.73, 1.21)	0.615	1.02 (0.78, 1.33)	0.879
	2019 vs. 2020	0.78 (0.60, 1.00)	0.051	0.79 (0.60, 1.03)	0.076
Meeting sleep recommendations	2016 vs. 2017	0.87 (0.69, 1.10)	0.240	0.89 (0.70, 1.13)	0.333
	2016 vs. 2019	0.62 (0.48, 0.81)	0.000	0.63 (0.48, 0.82)	0.001
	2016 vs. 2020	0.87 (0.70, 1.09)	0.222	0.89 (0.71, 1.13)	0.345
	2017 vs. 2019	0.72 (0.53, 0.98)	0.034	0.71 (0.52, 0.96)	0.028
	2017 vs. 2020	1.00 (0.76, 1.32)	0.982	1.01 (0.76, 1.33)	0.952
	2019 vs. 2020	1.40 (1.04, 1.88)	0.026	1.43 (1.06, 1.93)	0.020
Low–moderate risk on Kessler scale	2016 vs. 2017	1.92 (1.55, 2.38)	0.000	1.91 (1.52, 2.38)	0.000
	2016 vs. 2019	2.12 (1.70, 2.64)	0.000	2.12 (1.69, 2.66)	0.000
	2016 vs. 2020	2.24 (1.83, 2.74)	0.000	2.31 (1.86, 2.86)	0.000
	2017 vs. 2019	1.10 (0.85, 1.43)	0.467	1.11 (0.85, 1.45)	0.434
	2017 vs. 2020	1.17 (0.91, 1.49)	0.226	1.21 (0.94, 1.56)	0.138
	2019 vs. 2020	1.06 (0.82, 1.36)	0.661	1.09 (0.84, 1.41)	0.513

## Data Availability

Data are available on reasonable request to the authors.

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
