# Peer review of "Trends in Health-Risk Behaviors and Psychological Distress among Australian First-Year University Students"

_ijerph, 2024, doi:10.3390/ijerph21050620_

Round 1

Reviewer 1 Report

Comments and Suggestions for Authors

The paper aims to explore trends in health risk behaviors (poor diet, physical inactivity, sedentary behaviors, alcohol intake, smoking, drug use, and poor sleep) and psychological distress among first year Australian university students over four survey years from 2016 to 2020.

Major points:

Mostly descriptive results are presented. Despite the multitude of individual factors included in the study and described in the paper, the effect of none of them has been explored. In this form, the paper does not contribute significantly to the literature.

Significant gender differences are known to exist in regard to mental wellbeing. Gender differences exist also in regard to some health behaviors. Not exploring/controlling for and discussing the effects of gender is a major flaw.

What is the point of the very detailed description of all demographic information included in the study in Table 2? The role of factors such as hours of paid work, living situation and financial support is not discussed in the paper, they are irrelevant to its aims.

Table 3 should be replaced by a figure illustrating visually the prevalences in the 4 time points, i.e. the trends. It is nearly impossible to follow trends based on Table 3.

No rationale is provided for excluding first year students aged 25 or more years. If it is based on a developmental/life stage theory, it should be integrated into the paper from introduction to discussion.

Phrases like “high prevalence of health risk behaviors” and “health behaviors were generally poor” are used too loosely and non-specifically (it should be defined what “high” is). The prevalences of some health behaviors cited as found in other studies or found in your own study are not high. Some prevalences (e.g. smoking behavior) are actually very low, especially when compared to other parts of the world.

Minor points:

“Takeaway” should be replaced by “fast food” (which is what has been studied, there are also healthy options to take away).

Mistake in Table 3: data included on “Drug use in previous 3 months” which is not analyzed in the paper.

Statistical data included in Table 4 is then repeated in the text (there should be no repetition).

Reviewer 2 Report

Comments and Suggestions for Authors

The research is valuable in terms of determining the physical and mental health problems, and monitoring them the change over time of students who are just starting university and in the transition period from adolescence to young adulthood. Unfortunately, I believe that the study results are not sufficient and reliable due to some mistakes and shortcomings made in the study design at the beginning. The authors said in the introduction section that this study aims to investigate trends in health risk behaviors (poor diet, physical inactivity, sedentary behaviors, alcohol intake, smoking, drug use and poor sleep) and psychological distress among Australian university students aged 17 to 24 years, over four survey years from 2016 to 2020. But in the method section, it is observed that only the first-grade students were taken to study. Then why was a mail survey sent to all students? Although the researchers planned a cross-sectional study, they did not calculate a minimum sample size. Therefore, it is also impossible to talk about the power of the study. In addition, it was not stated how many first-graders there are for each research year and how many of them answered the survey. It also has not been specified how many first-graders there were in each research year and the proportion of respondents to the survey. The gender ratios of first-graders are not specified. However, the fact that more than 70% of the students in the sample are female students does not seem to represent the entire university. The fact that only volunteers participated in the survey and were encouraged by monetary rewards is another reason for bias in sample selection. Another question mark is that it has not been explained why data was not collected in 2018. In order to compare student behavior and health conditions between the years, it would be more appropriate to conduct multivariate and trend analyses instead of univariate analyses. Finally, the year 2020 was a year in which face-to-face education is suspended due to the pandemic and the whole world is affected both physically and emotionally. Therefore, consequences of the 2020 survey require to be considered from a different point.
